# Roles of the Core Components of the Mammalian miRISC in Chromatin Biology

**DOI:** 10.3390/genes13030414

**Published:** 2022-02-24

**Authors:** Gaspare La Rocca, Vincenzo Cavalieri

**Affiliations:** 1Cancer Biology and Genetics Program, Memorial Sloan Kettering Cancer Center, New York, NY 10065, USA; 2Department of Biological, Chemical and Pharmaceutical Sciences and Technologies, University of Palermo, 90128 Palermo, Italy

**Keywords:** miRISC, Argonaute, microRNA, TNRC6, chromatin, epigenetics

## Abstract

The Argonaute (AGO) and the Trinucleotide Repeat Containing 6 (TNRC6) family proteins are the core components of the mammalian microRNA-induced silencing complex (miRISC), the machinery that mediates microRNA function in the cytoplasm. The cytoplasmic miRISC-mediated post-transcriptional gene repression has been established as the canonical mechanism through which AGO and TNRC6 proteins operate. However, growing evidence points towards an additional mechanism through which AGO and TNRC6 regulate gene expression in the nucleus. While several mechanisms through which miRISC components function in the nucleus have been described, in this review we aim to summarize the major findings that have shed light on the role of AGO and TNRC6 in mammalian chromatin biology and on the implications these novel mechanisms may have in our understanding of regulating gene expression.

## 1. Introduction

Most of the mammalian genome is transcribed into non-coding RNAs (ncRNAs) that play a variety of structural and regulatory roles in cells. Small-size RNAs (sRNAs), ranging from ~18 to 30 nucleotides, are a well-known class of ncRNAs involved in both the cytoplasmic and nuclear regulation of gene functions. Their primary sequence provides the information necessary to guide repressor or activator complexes to complementary binding sites within a target nucleic acid. The mode of recognition of nucleic acids by sRNAs exploits the pairing combinations provided by the Watson–Crick complementarity. This leads to a higher degree of binding specificity than that allowed by proteins, which instead rely on a limited number of binding domains to recognize DNA and RNA sequences.

Mammalian genomes code for three main groups of sRNAs: microRNAs (miRs) [1], endogenous small interfering RNAs (endo-siRNAs) [2], and PIWI-interacting RNAs (piRNAs) [3]. Although their genomic origin, length, function, and expression pattern among tissues varies, these three groups of sRNA share one unifying feature: in their search for target sites, they are physically associated with one member of the Argonaute protein family. Based on amino acid sequence similarities, the mammalian Argonaute family can be classified into two clades: the Argonaute (AGO) clade, represented by the four paralogues AGO1-4 (reviewed in [4]), and the P-element-induced wimpy testis (PIWI) clade [5]. Each sRNA class selectively associates with one specific family of Argonaute proteins. piRNAs associate with PIWI proteins, while miRs and endo-siRNAs associate with AGOs. In each of these complexes, the role of Argonaute is to promote the proper steric conditions to favor base pairing between the sRNA and the target site [6], thereby enabling sRNA-mediated regulatory processes (reviewed in [4]). PIWI proteins and their associated piRNAs are mainly expressed in the germline, where they are involved in the silencing of transposable elements. Mechanisms underlying the piRNA pathway will be briefly discussed in the next section, but we direct the reader elsewhere for a thorough review on the topic [5].

In contrast, AGOs are expressed ubiquitously and show a remarkable versatility in function. This is supported by the fact that AGOs are embedded in a variety of complexes that range from 100kD to several MD [7,8], implying an intrinsic ability of this protein to bind to different partners. Moreover, mechanisms of shuttling between different subcellular compartments have been described [8], highlighting the potential of AGOs to act in a multiplicity of roles in different subcellular contexts (reviewed in [9]). 

The most studied among AGO functions in mammalian cells is its miR-dependent post-transcriptional gene repression (PTGR) of protein-coding genes [1]. miRs were initially discovered in *Caenorhabditis elegans (C. elegans)* as heterochronic genes [10]. Since then, subsequent studies have established that miRs’ regulated expression is key to enable timely mRNA expression during development and physiology in most eukaryotes. The newly released miRbase 22.1, the most comprehensive database cataloging all annotated miRs so far, indicates that about 2600 miRs originate from the human genome [11].

During PTGR, AGO is directed onto target mRNAs via base pairing between a 7 to 8nt “*seed*” sequence within the loaded miR, and a binding site usually situated within the 3’UTR or the coding region of the cognate target mRNA. Next, the AGO–miR complex associates with one member of metazoan-specific TNRC6 proteins, a molecular docking platform for the recruitment of mRNA decay enzymes, namely decapping factors and deadenylases [12] (Figure 1).

Therefore, AGO, the miR, and TNRC6 constitute the core of the miR-induced silencing complex (miRISC) [17], the effector machinery that mediates PTGR in the cytoplasm. miRISC-mediated PTGR accounts for the repression of more than half of metazoan protein-coding genes [18], representing a layer of post-transcriptional gene repression involved in virtually all cellular processes. As AGO and TNRC6 are utilized prevalently in the miRISC-mediated PTGR, they are enriched in the cytoplasm in most cell types. However, they can also be found in the nucleus [19,20,21], and can become further enriched in this compartment under specific conditions, such as during stress [22].

It has been demonstrated that multiple molecular pathways allow for the controlled shuttling of AGO and TNRC6 between the nucleus and cytoplasm. Nuclear import of AGO is controlled, at least in part, by Importin 8 [23], although it is conceivable that different routes of nuclear import/export with redundant functions may be involved. For example, AGO nuclear import has also been shown to be mediated by Importin-beta, and its export by Exportin-1/CRM1 in a RAN-dependent manner [24,25].

Similarly, TNRC6 proteins can shuttle between the cytoplasm and the nucleus, which is supported by the fact that some TNRC6 members contain both a nuclear localization signal (NLS) and nuclear export signal (NES). Accordingly, TNRC6 export has been shown to be mediated by the interaction of its NES with Exportin-1 [26].

As the physical interaction between AGO and TNRC6 seems to be conserved between the nucleus and cytoplasm, [27] it is not surprising that the myriad of regulatory pathways that control the localization of one protein would inevitably affect the localization of the other, and that a stable binding of the AGO–TNRC6 complex to the target mRNA is required for this complex to endure in the cytoplasm [24,26].

Although AGO–TNRC6 complexes can still mediate PTGS on target mRNAs before they reach the cytoplasm [28], they can also mediate diverse nuclear-specific functions [9,29]. Among these nuclear-specific functions, the processes where AGO–TNRC6 complexes cooperate with the chromatin modifying and remodeling machineries to control gene expression at the epigenetic level, will be the focus of this review. Although we will focus on the epigenetic mechanisms of transcriptional gene silencing (TGS), and transcriptional gene activation (TGA) in mammals, we will also briefly discuss homologous regulatory processes in yeast and lower animals to highlight the conservation of these mechanisms through evolution. In particular, we will mention the piRNA pathway, where the PIWI clade of Argonautes mediates the silencing of transposons in the germline through well-characterized epigenetic mechanisms similar to those mediated by the AGOs in the soma.

## 2. Evolutionary Conservation of AGO-Mediated TGS Mechanisms

The isolation of the RITS in *Schizosaccharomyces pombe S**. pombe*). From yeast to humans, heterochromatin constitutes a group of transcriptionally silent chromatin domains essential for the maintenance of chromosome integrity and for the regulation of gene expression [30]. Heterochromatin regions are marked by the presence of members of the heterochromatin protein 1 (HP1) family that bind to repressive histone marks, such as di- and trimethylated histone H3 lysine 9 (H3K9me2 and H3K9me3, respectively) [31] through a conserved N-terminal domain named the *chromodomain*. H3K9 methylation spreads along chromatin through sequential cycles of methylation, mediated by H3K9 methyltransferase enzymes, coupled to the oligomerization of HP1 family members [32]. A link between AGO proteins and heterochromatin remodeling has been extensively characterized in *S. pombe.* The mechanistic basis of this link began to be elucidated with the isolation of the RNA-induced transcriptional silencing (RITS) nuclear complex, whose core components are Chp1 (a chromodomain-containing protein), Ago1, a Dicer-dependent sRNA, and Tas3 [33], a protein that contains a GW-rich domain with “AGO-hook” properties [34]. The RITS was shown to mediate heterochromatin formation at centromeric and pericentromeric DNA sequences complementary to the Ago1-associated sRNAs through TGS (Figure 2A). Interestingly, the sRNAs loaded on the RITS, often referred to as repeat-associated siRNAs (rasiRNAs) [35], are produced by the Dicer-mediated processing of long dsRNAs that results from convergent antisense transcription of the same centromeric regions where heterochromatin assembly initiated [36]. Deletion of any of the RITS component genes, or of *dicer* with the consequent loss of rasiRNAs, induces defects in heterochromatin maintenance. Once bound to the chromatin, the RITS recruits the RNA-dependent RNA polymerase complex (RDRC) formed by Rdp1 polymerase, the helicase Hrr1, and the non-canonical poly(A) polymerase Cid12. The RDRC complex amplifies the silencing process initiated by RITS in a positive feedback loop consisting of the further synthesis of target-specific dsRNA, their processing by Dicer and subsequent production of rasiRNA, and additional recruitment of more RITS to the region [37] (Figure 2A).

Although the RITS can, in principle, recognize DNA sequences to target specific chromatin regions, several studies have supported a model according to which the sequences within a transcript generated from the targeted locus act instead as the docking site for the rasiRNA-loaded Ago1 [37,39,40,41] (Figure 2A).

In addition to the nascent transcript and the RDRC and RITS complexes, the heterochromatin assembly platform is also comprised of the CLRC complex, which is required for H3K9-specific methylation and heterochromatin maintenance. The core components of CLRC are the histone methyltransferase (HMT) Clr4, the Cullin scaffold protein Cul4, the RING finger protein Pip1, the DNA damage binding protein 1 homolog Rik1, and the DCAF-like protein delocalization of Swi6 1 and 2 (Dos1 and Dos2) [42,43,44,45,46]. A LIM domain protein, Stc1, mediates the interaction between CLRC and the RITS complexes, and is required for RNA-dependent H3K9 methylation [47]. The RITS, RDRC, and CLRC complexes physically interact, and conceivably work in synergy to establish and maintain heterochromatin structure.

The mechanism underlying Ago-mediated heterochromatic modifications in plants resembles the mechanism in *S. pombe*. For example, in *Arabidopsis thaliana*, the DICER-LIKE proteins DCL1 to 4 produce sRNAs that are loaded into 1 of the 10 Argonaute proteins encoded by the genome of this organism. Of note, in plants, sRNA-induced TGS relies principally on DNA methylation at cytosine residues by the RNA-directed DNA methylation pathway (RdDM). RdDM directs de novo DNA methylation while other DNA methyltransferases, such as MET1 or CMT3, mediate DNA methylation maintenance [48]. 

The piRNA pathway. Mechanisms of TGS are also present in animals. TGS in animals is best understood in the piRNA pathway, where the Piwi clade of Argonautes direct the silencing of transposable elements (TE) in gonadal tissues [3,49]. In *Drosophila*, Piwi proteins/piRNA complexes operate in the nucleus where they recognize nascent transposon transcripts and direct the TGS of the corresponding transposon loci [50,51]. Recognition of the nascent transcript by Piwi–piRNA complexes is stabilized by the Piwi-interacting zinc-finger protein Asterix (Arx) [52]. Panoramix (Panx), a protein with no known domain [52,53,54], is also a key TGS effector, which acts at the interface between the piRNA pathway and the chromatin silencing machinery [51,54]. Downstream of Arx and Panx, the coordinated action of dLsd1/Su(var)3–3 and Eggless/dSETDB1 leads to the removal of H3K4me2 and to the establishment of H3K9me3, respectively, followed by heterochromatin formation via the recruitment of HP1a/Su(var)205 [49,55,56,57] (Figure 2B).

Similar mechanisms govern TGS in *C. elegans*. In this organism, sRNAs named 22G-RNAs repress transposable elements in the germline [58,59]. The Argonaute protein HRDE-1/WAGO-9, loaded with 22G-RNAs, shuttles to the nucleus and recognizes nascent RNA transcripts. The coordinated action of additional factors, such as NRDE-1, -2, and -4, results in RNA pol II stalling, H3K9me3 deposition, and the binding of HPL-2—a HP1a homolog—ultimately resulting in the transcriptional silencing of the target locus [60,61] (Figure 2C). 

Although the piRNA pathway has been primarily described as a mechanism to repress transposons, a large portion of identified piRNAs in *C. elegans* does not overlap with known transposon sequences [62], suggesting other possible regulatory functions mediated by the piRNA pathway. Growing evidence shows that piRNAs are indeed involved in the global regulation of endogenous transcriptional programs. For example, Cornes et al. have recently reported that piRNAs mediate the transcriptional silencing of endogenous genes to promote sperm differentiation [63].

As described above, TGS mechanisms in evolutionarily distant species share common mechanisms relying on: (1) the recognition of nascent transcripts by an Argonaute family loaded with sRNA; (2) the recruitment of heterochromatin factors; and (3) silencing of the targeted loci. As described in the next section, similar mechanisms also take place in mammals.

## 3. Evidence of TGS in Mammals

TGS triggered by exogenous sRNAs. Several examples of TGS in mammals have been described (Table 1). Early studies in cell lines have shown that sRNAs that are exogenously delivered could inhibit gene expression and modify chromatin [64,65,66,67]. For example, synthetic sRNAs that are complementary to sequences within the elongation factor 1 alpha (*EF1A*) promoter could repress the expression of an integrated *EF1A* promoter–reporter transgene, and, although to a lesser extent, the expression of the endogenous *EF1A* locus [65]. In this model, gene silencing was associated with cytosine DNA methylation (henceforth DNA methylation, a hallmark of silent chromatin), which was abolished by treatment with 5-azacytidine, an inhibitor of DNA methyltransferases, indicating that gene silencing occurred entirely through a mechanisms similar to the TGS described in *S. pombe,* and did not involve RNA-mediated PTGR events [65].

However, it became clear that DNA methylation was not always associated with sRNA-mediated TGS [67,68,69]. For example, the absence of DNA methylation during TGS triggered by exogenous sRNAs was also observed in the case of the *progesterone receptor* (*PR*) gene [69]. Ting at al. have shown that, although sRNAs directed to the *E-cadherin* (*CDH1*) promoter could promptly induce transcriptional repression, this process could take place even in cells depleted of virtually any capacity to perform DNA methylation [67].

While the extent to which DNA methylation participates in sRNA-induced TGS is still under debate, more compelling evidence has been provided for sRNA-induced silencing through repressive histone modifications [65,70]. For example, Weinberg et al. showed a fourfold increase in H3K9 and H3K27 methylation in proximity of the endogenous *EF1A* promoter upon transfection of promoter-directed sRNAs. Interestingly, H3K9 methylation was even induced at sequences hundreds of bps downstream of the *EF1A* promoter, suggesting that the sRNA-mediated targeting may induce the spreading of chromatin marks to regulate neighboring regions [70].

The observation that promoter-directed sRNAs could induce chromatin changes in mammalian cells ignited interest towards the factors that could mediate this process. The structural and functional information previously gathered from the RITS and the piRNA pathways in *Drosophila* and *C. elegans* inspired the search for a potential role of AGO as the physical interface between sRNAs and the chromatin-modifying machinery. Such a role for AGO came to light in chromatin immunoprecipitation (ChIP) experiments in human cells. These studies showed that the intracellular delivery of sRNA could induce up to a 20-fold increase of AGO1 and H3K9me2 presence in proximity of the sequences complementary to the sRNA [71]. Interestingly, H3K9me2 was detectable up to 300 bp downstream of the sRNA-binding site, suggesting that the AGO–sRNA complex, by recruiting histone methyltransferases (HMTs), initiates the formation of heterochromatic foci, which eventually propagate to the surrounding regions. AGO1-directed TGS and H3K9me2 formation required protein–protein interactions between AGO1 and both TRBP2 and Pol II, and was accompanied by an enrichment of the HMT EZH2, a Polycomb component responsible for the synthesis of H3K27me3. Within this multi-subunit complex, the role of the TRBP2 is not fully understood, although it may mediate the loading of the guide sRNA on AGO, similar to its role in the miRLC in the cytoplasm [72]. The recruitment of EZH2 onto the sRNA-targeted site may be necessary for H3K27me3 accumulation, linking AGOs to the endogenous mechanisms of Polycomb silencing. Finally, the requirement of Pol II is consistent with the formation of a nascent transcript that acts as the landing platform for the AGO1–sRNA complex, although the role of such a transcript was not explored in detail in this early study.

Recent studies have further elucidated the identity and modus operandi of the enzymatic activities recruited by AGO during TGS, although the high complexity of mammalian genomes has made the search for these enzymes more challenging than the search previously undertaken in *S. pombe*. For example, while Clr4 is the only H3K9-specific methyltransferase within the RITS in *S. pombe*, mammalian genomes encode for several HMTs with H3K9 specificity, any of which could, in principle, function during RNA-induced TGS. So far, among these enzymes a role in TGS has only been documented for SETDB1, as evidenced in a study in which sRNAs directed to the *androgen receptor* (*AR*) promoter increased the local recruitment of both AGO2 and SETDB1 around the sRNA binding site [73]. Moreover, EZH2 was also enriched on the targeted promoter and was physically associated with SETDB1, suggesting that the two complexes may cooperate during TGS in mammalian cells. SIN3A and HDAC2, key members of the SIN3–HDAC corepressor complex, whose function is to promote histone deacetylation and heterochromatin formation, are also enriched at the targeted promoter in a AGO2- and SETDB1-dependent manner and physically interact with SETDB1 [73]. These observations are consistent with a model where the SIN3–HDAC corepressor complex cooperates with AGO2 and SETDB1 in the establishment of silent chromatin in regions surrounding the sRNA target site. Importantly, this study showed that the recruitment of SETDB1–AGO2 and the formation of repressive histone marks within the region surrounding the *AR* promoter required a nascent ncRNA spanning the promoter region. This observation provides further evidence that a nascent RNA transcript may be the preferential recognition platform for sRNA-guided chromatin remodeling of the template DNA. Notably, DNA methylation is not induced following targeting of the *AR* promoter, consistent with the notion that the induction of repressive histone modifications may play a greater role in mammalian TGS.

miR-dependent pathways of TGS. The discovery that exogenous sRNAs could induce TGS and chromatin silencing in mammalian cells led to the speculation that endogenous sRNAs with analogous function may exist. miRs were an obvious class of candidates to explore, given their similarity in size with the RITS-associated sRNAs. Early attempts to discover an example of miRs-mediated TGS relied on computational tools that search for sequences within gene promoters with extensive complementarity to endogenous miRs. This approach led to numerous examples of miR-induced TGS in mammals (Table 1). Authors of the first report of miR-mediated TGS in mammalian cells showed that miR-320 is generated within the promoter of the cell cycle gene *POLR3D*, and its depletion induced an increase of *POLR3D* expression at the transcriptional level [74]. Moreover, the overexpression of miR-320 in HEK-293 cells induced an enrichment of AGO1, EZH2, and H3K27me3 onto the *POLR3D* promoter, consistent with a role of miR-320 in the induction of repressive chromatin at the targeted site. Sense transcripts spanning the *POLR3D* promoter were also detected, although authors have not investigated their role in AGO1 recruitment and gene repression. Given the central role of *POLRD3* in the cell cycle progression [75], this study was the first to highlight the potential implication of AGO-mediated TGS in proliferative disorders.

Using computational methods to screen for miR target sites within gene promoters, Younger et al. found multiple miRs that were able to target the promoter of the human *progesterone receptor* (*PR*) and of *immunoglobulin superfamily member 1* (*IGSF1*). Among those, the overexpression of miR-423-5p was shown to decrease Pol II occupancy, and to induce accumulation of H3K9me2 at the promoters of the two mentioned genes, in a DNA-methylation-independent manner. Moreover, TGS required the recruitment of AGO2 onto a ncRNA-encompassing gene promoter [76]. Of note, some studies using exogenous sRNAs directed to the *PR* promoter have ruled out a role for TNRC6 in the TGS of this locus [77].

More recently, Di Mauro et al. have found a link between Wnt signaling and miR-mediated epigenetic regulation in cardiac cells [78]. They have found that, upon drug-elicited inhibition of the Wnt pathway, the level of the cardiac-enriched miR-133a increases in the nucleus of HL-1 cardiac cells. Importantly, a knock-down of AGO2 (but not AGO1) and IMPORTIN-8 abrogated re-localization, suggesting a role of these two proteins in the shuttling of miR-133a into the nucleus. Enrichment of miR-133a in the nucleus coincided with the downregulation of several transcription factors and epigenetic enzymes, including *DNMT3b*, whose promoter contains a binding site for miR-133a. These authors were able to directly link the nuclear enrichment of miR-133a with the transcriptional silencing of *DNMT3*, as the transfection of cells with antisense oligonucleotides against miR-133a binding sites prevented *DNMT3b* repression. ChIP analysis showed that the inhibition of Wnt signaling was also accompanied by AGO2 localization at the *wnt* promoter, an increase in the histone repressive mark H3K27me3, and a decrease of H3K4me3 that marks active transcription. Interestingly, DNMT3B was also recruited at the miR-133a binding site and was physically associated with AGO2. As a consequence of the recruitment of DNMT3B, DNA methylation levels at the promoter increased, forming a negative feedback mechanism through which the miR-mediated recruitment of DNMT3B resulted in the repression of its own expression.

**Table 1 genes-13-00414-t001:** Summary of studies showing genes regulated by TGS mechanisms in mammals. The region targeted by the small RNA falls within the promoter of the gene, unless stated otherwise.

Targeted Gene	Targeting sRNA	RISC Component Involved	Resulting Epigenetic Marks at Target Site	Proposed Targeting Model	References
EF1A	Exogenous	undetermined	↑ DNA methylation	undetermined	[65]
RASSF1A	Exogenous	undetermined	↑ DNA methylation	undetermined	[64]
CDH1	Exogenous	undetermined	Absence of DNA methylation confirmed	undetermined	[67]
PR	Exogenous	undetermined	Absence of DNA methylation confirmed	undetermined	[69]
EF1A	Exogenous	undetermined	↑ H3K9 H3K27 methylation	undetermined	[70]
POLR3D	Endogenous miR-320	AGO1	↑ EZH2, ↑ H3K27me3	Nascent RNA	[74]
HOXB4/ HOXD4	Endogenous miR-10a/b	AGO1/3	↑ H3K27me3 ↑ DNA methylation	Nascent RNA	[79]
TBCEL	Endogenous miR-17-5p miR-20a	AGO1/2	↑ H3K9me2	Nascent RNA	[20]
RASA2	Endogenous miR-17-5p miR-20a	AGO1/2	↑ H3K9me2	Nascent RNA	[20]
RHPN2	Endogenous miR-17-5p miR-20a	AGO1/2	↑ H3K9me2	Nascent RNA	[20]
WHSC1	Endogenous miR-17-5p miR-20a	AGO1/2	↑ H3K9me2	Nascent RNA	[20]
PR	Endogenous miR-423-5p	AGO2	↓ Pol II ↑ H3K9me2	Nascent RNA	[76]
IGSF1	Endogenous miR-423-5p	AGO2	↓ Pol II ↑ H3K9me2	Nascent RNA	[76]
CDC2 CDCA8	Endogenous let-7f	AGO2	↑ H3K27me3 ↑ H3K9me2 ↓ H3K4me3	undetermined	[80]
DNMT3b	Endogenous miR-133a	AGO2	↑ H3K27me3 ↓ H3K4me3	undetermined	[78]
MMP-14	Endogenous miR-584-3p	AGO2	↑ H3K27me ↑H3K9me2	DNA	[81]
AR	Exogenous	AGO2	↑H3K27me ↑H3K9me2	Nascent RNA	[73]
Beta-actin (Termination region)	undetermined	AGO1/2	↑ H3K9me2 ↑ HP1	Antisense transcripts	[82]

↑ = Increase, ↓ = Decrease.

Endogenous miRs have also been involved in the transcriptional repression of the metastasis-promoting gene *matrix metalloproteinase 14* (*MMP-14*). This gene is activated by the transcription factor Yin Yang 1 (YY1), and its overexpression or downregulation promotes or inhibits tumor progression, respectively. Zheng et al. noted that a YY1 binding site is located just 30–40 bp downstream of a putative binding site for miR-584-3p within the *MMP-14* promoter, suggesting a possible functional interaction between the two elements. Indeed, in gastric cancer cells, the authors found that miR-584-3p binds to the *MMP-14* promoter in an AGO2-dependent manner, leading to increased levels of the repressive histone marks H3K27me3 and H3K9me2. The induction of heterochromatin hinders the recruitment of YY1 and consequently suppresses *MMP-14* expression. Importantly, treatment with RNase H reversed miR-584-3p-mediated chromatin modification, indicating that miR-584-3p directly interacted with the *MMP-14* promoter via an RNA–DNA hybrid. Authors of this study showed that, by impeding the binding of YY1 to the *MMP-14* promoter, an AGO2–miR-584-3p complex can suppress the tumorigenesis and aggressiveness of gastric cancer cells, both in cell lines and xenograft tumors [81].

It is unclear how widespread miR-mediated TGS occurs within mammalian gene networks, and to what extent either DNA or RNA acts as a recruitment platform. Nevertheless, the expression of ncRNA-spanning promoter regions seems to be a common feature of several mammalian promoters [83,84], which makes them potential targets of miR-mediated TGS mechanisms. The repurposing of target prediction algorithms to scan such promoter sequences for sites complementary to miRs is certainly a valid approach to potentially predict new examples of miR-mediated TGS [85,86].

## 4. The Role of AGO and TNRC6 Protein Families in sRNA-Dependent Transcriptional Gene Activation in Mammals

Beyond their typical role in the repression of gene expression, complexes that include sRNAs, AGO proteins, and, in isolated cases, TNRC6, have also been shown to promote gene expression via a mechanism known as transcriptional gene activation (TGA) (Table 2) (reviewed in [87]).

The first evidence of AGO-mediated TGA came from experiments where sRNAs, named small activating RNAs (saRNAs), targeting the promoter region of *CDH1, p21WAF1/CIP1* (*p21*), and *VEGF*, surprisingly stimulated, instead of inhibited, gene transcription in a sequence-specific manner. This process required AGO2 and was accompanied by the loss of the repressive mark H3K9me2 [88]. Notably, previous studies have shown that the *CDH1* promoter was susceptible to TGS via the induction of histone methylation [67]. Therefore, determinants must exist that would allow promoter-directed sRNAs to activate the expression of a gene in one context and inhibit it in another. The authors speculated that this apparent discrepancy may be due to the different levels of free energy involved in the two targeting events—TGS was induced by sRNA directed toward CpG islands, while TGA by saRNAs were directed toward promoter sequences with low GC content—which may eventually result in the recruitment of different complexes with opposite effects on gene expression [88].

TGA was further confirmed in studies where saRNA that was directed to the promoter of the *PR* gene increased the expression of the locus. ChIP analyses showed that TGA was accompanied by the reduced acetylation of H3K9 and H3K14, reduced presence of HP1, and increased levels of H3K4me2 and H3K4me3. Importantly, the recruitment of the AGO–saRNA complex required a nascent antisense RNA transcript overlapping the promoter [89,90]. However, Meng et al. have later found compelling evidence that saRNAs can activate the *PR* gene transcription via the recognition of DNA, rather than of a nascent transcript [91]. To support their model, they conducted a saRNA-mediated knock-down of all transcripts overlapping the *PR* promoter and observed no impairment of TGA at this region, thereby arguing against a possible role of nascent transcripts in this instance. As evidenced by the conflicting observations from these studies, the modality of target recognition during TGA is still an object of intense debate. As noted earlier, sRNAs directed to the PR promoter can also repress gene expression [76]. Although the mechanisms that would explain this specificity are still unknown, one possible explanation is that basal levels of expression of the PR gene may determine whether activation or silencing will occur upon sRNA binding [90].

**Table 2 genes-13-00414-t002:** Summary of studies showing genes regulated by TGA mechanisms in mammals. In the listed examples, the region targeted by the small RNA falls within the promoter of the gene, unless stated otherwise.

Targeted Gene	Targeting sRNA	RISC Component Involved	Resulting Epigenetic Event at Target Site	Proposed Targeting Model	References
*CDH1* *VEGF*	Exogenous	AGO2	↓ H3m2K4 ↓ H3m2K9	undetermined	[88]
PR	Exogenous	undetermined	↓ H3K9 ↓ H3K14 ↑ H3K4m2 ↑ H3K4m3	Nascent RNA	[89,90]
PR (distal site)	Exogenous	AGO2	undetermined	Genomic DNA	[91]
CDH1 CSDC2	Endogenous miR-373	undetermined	undetermined	undetermined	[92]
*CCNB1*	Endogenous miR-744 miR-1186	AGO1	↑ Pol II ↑ H3K4me3	undetermined	[93]
*IL24* *IL32*	Endogenous miR-205	undetermined	↑ Pol II ↑ H3K4me2 ↑ H3ac ↑ H4ac	undetermined	[94]
*HPSE*	Endogenous miR-558	AGO1	↓ H3K9me2 ↓ H3K27me3 ↑ H3K4me3 ↑ Pol II	Genomic DNA	[95]
FBP1 FANCC (enhancers)	Endogenous miR-24-1	AGO2	↑ Pol II ↑ H3K27ac ↑ H3K4me ↓ H3K9me3	undetermined	[96]
COX-2	miR-589 mimics	AGO2 TNRC6	↑ Pol II ↑ H3K4me3 ↑ H4Ac	Nascent RNA	[97]
p21	Exogenous	AGO2	↑ H2B ubiquitination	undetermined	[88,98]
Foxo3	Endogenous miR-195-5p	AGO2	undetermined	Genomic DNA	[99]
MyoD (enhancer)	?	AGO1	↑ H3K27ac	Enhancer RNA	[100]

↑ = Increase, ↓ = decrease.

## 5. miR-Dependent Endogenous Pathways of TGA in Mammals 

TGA mechanisms act through a promoter and enhancer. The first evidence that endogenous miRs can induce TGA came from a study in which the transcription of the *CDH1* and *CSDC2* genes, whose promoters contain sequences complementary to miR-373, was upregulated by the overexpression of miR-373. This early study, although it provided evidence for endogenous pathways that mediate TGA, did not investigate the potential involvement of AGO proteins or nascent transcripts in this process [92].

The mechanistic details on miR-mediated TGA were soon provided by Huang at al. By using a combination of in silico and wet lab approaches, these authors found that miR-744 and miR-1186 can bind to complementary sites within the *CCNB1* promoter and subsequently enhance *CCNB1* expression in mouse cell lines. ChIP analysis revealed that the expression of miR-744 and miR-1186 induced the localization of AGO1 on the *CCNB1* promoter, accompanied by increased levels of RNA Pol II and H3K4me3 at the *CCNB1* TSS [93]. Since *CCNB1* overexpression promotes tumorigenesis [101,102], and the overexpression of miRs miR-744 and miR-1186 phenocopies *CCNB1* overexpression, this work indicates that TGA may be a process through which miRs can contribute to cancer development.

Further studies have similarly supported a role of TGA in tumor formation and progression. Majid et al. showed that miR-205 binds to complementary sites within the promoters of the tumor suppressor genes *IL24* and *IL32*, enhancing their expression via TGA. This process led to induced apoptosis and cell cycle arrest, and impaired the invasive properties of prostate cancer cells. Accordingly, the overexpression of miR-205 was accompanied by an enrichment of Pol II, H3K4me2, and H3/H4ac in the promoter of both genes, indicating transcriptional activation [94]. However, although a role of AGO proteins in this instance is conceivable, this aspect was not formally addressed in this study.

By using neuroblastoma cells as a model, Qu et al. found that the overexpression of miR-558 induced the expression of the *heparanase* (*HPSE*) gene, which was accompanied by a corresponding decrease of H3K9me2 and H3K27me3, and an increase of H3K4me3 and RNA Pol II on the *HPSE* promoter. *HPSE* gene activation requires the presence of AGO1, but not AGO2, as only a knock-down of AGO1 abolished the miR-558-induced enrichment of active epigenetic markers on the promoter.

Although RNase-H treatment reduced TGA on the *HPSE* promoter, suggesting that the AGO1–miR-558 complex relies on DNA–RNA pairing for promoter recognition, such a mechanism has not been thoroughly investigated and no direct evidence has been provided for this model of targeting in this study [95].

miRs can mediate TGA by promoting a favorable chromatin status not only at the promoters, but also at the enhancers of target genes. Xiao et al. observed that about 25% of loci involved in the production of nuclear-enriched miRs overlap with markers of active enhancers, namely p300/CBP, H3K4me1, and H3K27ac, as well as Dnase I-sensitive regions. Moreover, they observed that the expression of this subgroup of miRs positively correlated with the expression of surrounding loci. These observations prompted them to test the hypothesis that miRs may target enhancers overlapping their own loci and, in turn, activate both their own expression and the expression of surrounding genes. This hypothesis was confirmed by ectopically overexpressing miR-24-1, whose locus maps within a region with enhancer function. Indeed, the overexpression of miR-24-1 led to the expression of the endogenous miR-24-1, as well as the expression of its neighbor protein-coding genes *FBP1* and *FANCC*. TALEN deletion of the enhancer blocked the ability of miR-24-1 to activate *FBP1* and *FANCC*. Importantly, TGA on this locus was abolished in a AGO2-null background, implying that miR-24-1-dependent TGA is mediated by AGO2. Moreover, miR-24-1 overexpression induced an enrichment of Pol II and the consequential transcription of RNAs overlapping the enhancer region, concomitant with increased levels of AGO2, p300, H3K27ac, and H3K4me1, as well as a reduced level of H3K9me3. Notably, the overexpression of miR24-1 not only led to an increased H3K27ac within the proximal enhancer, but also at thousands of enhancers with sequences complementary to the miR-24-1 [96].

It is also important to note here that, although promoters and enhancers seem to be the preferential target sites for TGA in mammals, the possibility that other genomic regions may also be recognized by AGO–miR complexes to initiate permissive chromatin has yet to be determined.

The purification of the RNA-induced transcriptional activation complex. Recently, a complex able to mediate TGA has been characterized. Using the human *CDKN1A (p21)* as a model gene, Portnoy et al. have isolated the RNA-induced transcriptional activation (RITA) complex, which, in addition to the saRNA–AGO2 complex, includes the nuclear DNA helicase II (RHA) and CTR9 [98]. RHA is known to bind to DNA and mediate transcriptional activation by recruiting basal transcription machinery to promoter DNA and/or modifying chromatin structure [103]. The presence of RHA in the RITS may suggest that the targeting of this promoter by saRNA involves the recognition of genomic DNA. However, this aspect is not clear, and the recognition of a nascent transcript may still be the preferential targeting mechanism in this context (Figure 3). CTR9 is a component of PAF1C, which is required to recruit histone-modifying factors such as E2/E3 ubiquitin ligase to the Pol II complex to mediate H2B ubiquitination, and, consequently, H3K4 and H3K79 methylation [104]. RITA also recruits RNA Pol II, leading to productive transcription elongation, which highlights its polyvalent role in activating gene expression. TNRC6 and components of the miRISC-loading complex (DICER, TRBP) were absent from the RITA complex, suggesting that the AGO2 pool employed in TGA associates with a unique set of proteins distinct from those found in the cytoplasmic miRISC. However, as described below, there are instances in which TNRC6 can associate with AGO proteins to mediate TGA in mammalian cells, leading to the speculation that its requirement in this process may be context- or gene-specific.

## 6. The Involvement of AGO in Chromatin Remodeling and Splicing

The building blocks of chromatin are the nucleosomes, which are composed of histone proteins wrapped by genomic DNA. Chromatin-remodeling complexes include enzymes that ensure the proper spacing between nucleosomes, which ultimately contribute to controlling the regulation of gene expression [105]. By performing a combination of SILAC-based proteomic and ChIP analyses in human cells, Carissimi et al. have found that AGO2 can interact with components of the SWI/SNF complexes, including BAF155, BRG1, and BAF47 [106]. SWI/SNF is a chromatin-associated machinery that controls the occupancy of nucleosomes onto DNA sequences, thereby affecting the access of DNA-binding proteins to regulatory sequences. The association between AGO2 and SWI/SNF was DNA- and RNA-independent, demonstrating a direct interaction between protein components.

Interestingly, the pool of SWI/SNF-associated AGO2 was associated with a group of DICER-dependent sRNAs that do not belong to any previously characterized sRNA class, and that, instead, preferentially map around SWI/SNF-bound TSSs. This class of sRNAs, named swiRNAs, was found to overlap with more than 4000 TSSs genome-wide, and co-localized with histone markers of open chromatin, such as H3K9ac and H2Az, suggesting widespread roles for swiRNAs within mammalian gene networks. As expected, micrococcal nuclease digestion (which detects open chromatin regions) followed by RNA-seq revealed that in an AGO2-null background, nucleosome occupancy at the +1 position of the targeted TSS is altered, indicating that AGO2 not only physically interacts, but also functionally cooperates, with SWI/SNF to control chromatin remodeling. However, the low expression of swiRNAs (they are about 4 orders of magnitude less expressed than miRs), in addition to the fact that the AGO2 ablation did not result in significant perturbation of expression of swiRNAs-targeted genes, indicates that more studies are needed to establish the precise role of this class of sRNA in vivo.

In 3D, chromosomes are organized into active and inactive territories [107], which are subdivided in topologically associated domains (TADs) [108,109]. A regulated 3D chromatin structure allows enhancers and promoters to interact spatially and functionally to regulate gene expression (reviewed in [110]). Emerging evidence points toward a role of AGOs in the control of enhancer–promoter interaction [111,112]. For example, Shuaib et al. have recently provided the first evidence of a role of AGO proteins in the regulation of chromatin 3D structures in mammalian cells [112]. By using a combination of genome-wide approaches, they showed that AGO1 binds to active chromatin regions with predicted enhancer function, as demonstrated by the presence of H3K4me1 and H3K27ac at these regions. Importantly, RNase treatment significantly reduced AGO1 protein level in the chromatin fraction, indicating that the presence of enhancer-associated transcripts is required for the recruitment of AGO1 to the targeted regions. AGO1 depletion resulted in the perturbation of the 3D genome architecture and led to a significant change in the expression of coding and non-coding genes located within 50kb of AGO1 binding sites. Although the mechanisms mediating this process are not fully elucidated, these observations support a role for enhancer-associated AGO1 in determining gene expression programs via the topological regulation of chromatin organization.

Mechanisms where AGO regulates alternative splicing via chromatin modification have been also described. Alló et al. have shown that exogenous sRNAs targeting both intronic and exonic regions of the fibronectin gene were able to regulate its alternative splicing by inducing an increase of H3K9me2 and H3K27me3, histone markers associated with gene silencing. This process was dependent on AGO1 and AGO2, and leads to a decrease of Pol II elongation, an event that ultimately affects splicing [113]. Similar AGO1- and AGO2-dependent control of alternative splicing via histone modification have also been described for the *CD44* gene [114]. Importantly, in a study aimed to assess the role of AGO1 in splicing regulation at the genome-wide level, Alló et al. showed that AGO1 depletion provoked changes in constitutive splicing for thousands of internal introns and in the patterns of ∼700 alternative splicing events, suggesting a widespread usage of this regulatory mechanism in the control of gene expression [115].

## 7. The Involvement of TNRC6 in TGA and Chromatin Modification

As discussed above, the role and contribution of AGO proteins in chromatin modification and organization has been amply demonstrated. In contrast, even though TNRC6 is associated with AGO also in the nucleus, evidence for its role in these nuclear-specific processes are scarce.

Nevertheless, TNRC6 proteins have been involved in TGA in some circumstances. Using the pro-inflammatory gene *COX-2* as a model, Matsui et al. provided the first evidence of a role of TNRC6 proteins in nuclear TGA through a mechanism mediated by endogenous miRs [97]. They found that miR-589 possesses seed sequence complementary to a sense RNA spanning the *COX-2* promoter, which led them to investigate the possibility that this miR may transcriptionally regulate *COX-2* expression. Indeed, anti-sense mediated knock-down of miR-589 or the introduction of miR-589 mimics in mammalian cells led to decreased, or increased, *COX-2* expression, respectively. Importantly, the knock-down of a promoter-associated transcript reduced the levels of COX-2 mRNA upon miR-589 overexpression, consistent with a role of promoter-associated RNAs as a docking platform for the recruitment of gene-activating machineries.

Upon the delivery of miR-589 mimics, *COX-2* induction was paralleled by the accumulation of AGO2 and RNA Pol II at the promoter. Importantly, an accumulation of TNRC6 at the same site was also observed. siRNAs directed against either AGO2 or TNRC6 reduced the ability of miR-589 mimics to induce *COX-2* expression, suggesting a functional role of these two proteins in the miR-dependent activation of this locus. Moreover, knocking down the sense strand of the promoter-associated RNA also resulted in the loss of miR-568-dependednt *COX-2* activation.

Together with the observation that AGO and TNRC6 physically interreact in the nucleus, these data are consistent with a model according to which AGO2 forms a complex with TNRC6 and miR-589, which, by recognizing nascent promoter-associated RNAs, induces the activation of the *COX-2* gene.

miR-589 mimics also induced the accumulation of WDR5, a protein scaffold that stimulates HMT activity [116], on the *COX-2* promoter. Accordingly, WDR5 recruitment was accompanied by the enrichment at the promoter of the activating histone markers H3K4me3 and H4Ac. Knocking down *WDR5* resulted in a complete loss of miR-mediated COX-2 activation. These observations are consistent with a role of AGO and TNRC6 proteins as a functional and physical interface between ncRNAs and chromatin-modifying complexes.

Building upon these observations, Hicks et al. aimed to isolate additional factors that may be involved in the intercommunication between AGO/TNRC6-containing complexes and the chromatin modifying machinery. To this aim, they performed a mass spectrometry analysis to identify TNRC6 partners in the nucleus [117]. In addition to the expected AGO proteins and the previously identified chromatin modifier WDR5, this analysis also found several other factors that bind to TNRC6 and that are putative mediators of chromatin modifications downstream of TNRC6. Among these factors were members of histone modifying complexes, such as MLL3, MLL4, ASH2L, RbBP5, NCOA6, NAT10, PCF11, GATAD2A, and ZNF24. Moreover, members of the SWI–SNF complex, ARID1A and SMARCD1, and members of the CCR4–NOT complex were also found to associate with TNRC6. The depletion of WDR5, NAT10, and MED14 abrogated TGA at the COX-2 locus, confirming their role in miR-mediated TGA at the *COX-2* locus. Other groups of TNRC6-associated proteins with potential chromatin-modifying functions were CCR4–NOT, which, in addition to its role in PTGR in the cytoplasm, has been shown to induce activating histone markers, including H3K4 tri-methylation, as well as H3 and H4 acetylation [118].

Although the miR-589/*COX-2* system represents a clear example of co-participation of AGO–TNRC6 complexes and chromatin-modifying factors, it also provides insights on the possible role of miRs in promoting chromatin 3D organization. This notion stems from the observation that all of the manipulations that affected the miR-589-dependent regulation of *COX-2* also perturbed the expression of the pro-inflammation gene *PLA2G4A*. *COX-2* and *PLA2G4A* loci map within the same chromosome 150kb apart, and are found in a head-to-head organization. Interestingly, an miR-589-dependent increase of the activating histone markers H3K4me3 and H4Ac within the *COX-2* promoter was accompanied by the same changes of histone markers spanning the *PLA2G4A* promoter. Importantly, no RNA was found to overlap both promoters, ruling out the possibility that miR-589 may simultaneously target *COX-2* and *PLA2G4A* through the same mechanism. Instead, chromosome conformation capture (3C) analysis revealed that miR-568 promotes a physical linkage between the two genes by inducing chromatin loops that place the two promoters in proximity to each other. Through this mechanism, the miR-dependent recruitment of AGO–TNRC6, with the consequent recruitment of chromatin-remodeling enzymes, may simultaneously induce histone modification on both the *COX-2* and *PLA2G4A* promoters. However, how prevalent this mechanism is in mammalian gene networks is yet to be determined.

## 8. Conclusions

Studies conducted over the past 15 years have established that in mammals, sRNAs can carry the information necessary to direct chromatin-remodeling machinery on specific genomic regions to regulate gene expression. However, the underlying mechanisms are still largely unknown, and many questions remain open. For example: (1) What are the determinants that drive either TGA or TGS on a given target gene? (2) What is the relative contribution of the AGO1-4 paralogues in the silencing or activating process? (3) Is AGO2’s slicing activity required? (4) Does the sRNA recognize the DNA sequence or a nascent transcript? (5) To what extent is DNA methylation involved in the silencing process? Like most biological processes, it is conceivable that context-specific conditions dictate the modality through which TGS and TGA function at a specific gene locus, which would explain the mechanistic complexity, and at times inconsistency, that emerges from the studies discussed in this review.

Another important question pertains to the requirement of TNRC6 paralogues in sRNA-mediated transcriptional regulation, as TGA of *COX-2* remains, so far, the only instance in which of TNRC6 seems to be involved. However, it is possible that the use of the AGO–TNRC6 dimer in TGA/TGS may be more widespread than currently appreciated. This idea came from the observation that the RITS, which plays a central role in TGS in *S. pombe* and contains a member of the Ago protein family, shares important structural similarities with mammalian TNRC6. Indeed, mammalian TNRC6 contains both the AGO-hook, a domain of Tas3, and an RNA-recognition motif (RRM), a part of Chp1 [119]. To date, however, the extent at which TNRC6 paralogues are used in transcriptional gene regulation is not known and more research is needed in this regard.

Our groups have recently generated a mouse model that allows the inducible disruption of the interaction between AGO and TNRC6. It is based on the expression of a small protein that competes with TNRC6 for binding to AGO, thereby interrupting PTGR without affecting the expression of miRs [120,121]. When coupled with NLS and associated import proteins, this system can, in principle, be used to disrupt the interaction between AGO and TNRC6 within complexes that mediate chromatin remodeling functions, and the consequent chromatin changes be assessed genome-wide to uncover genomic loci in which the AGO-TNRC6 interaction may be requited for chromatin regulation.

The ability to control gene expression by directing sRNAs to promoters and enhancers has motivated the research of possible approaches to transcriptionally adjust gene expression for therapeutical purposes [122]. For example, MiNA Therapeutics is one of the several biotech companies which have started to harness the mechanisms of sRNA-mediated gene regulation to develop anti-cancer therapies, and has recently initiated a phase II clinical trial to test sRNAs directed to CCAAT/enhancer-binding protein-α (CEBPA), a transcription factor that regulates myeloid cell development. As CEBPA expression is downregulated in myeloid cells, driving cells to an immunosuppressive phenotype, the activation of CEBPA may restore immune functions, thereby potentiating the effectiveness of co-administered anticancer drugs.

In conclusion, there is compelling evidence of the existence of mammalian mechanisms that utilize the information contained in sRNAs to direct transcriptional gene regulation via chromatin modification and remodeling. This information has been used to create new sRNA-based drugs to adjust gene expression, which, though still in their infancy, have opened a promising new field of therapeutical intervention, although more mechanistic studies are needed to improve the efficacy and safety of this new class of molecules.

## Figures and Tables

**Figure 1 genes-13-00414-f001:**
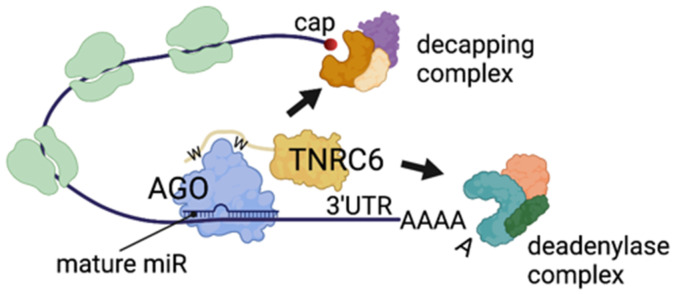
Mechanisms of PTGR mediated by AGO-containing complexes. A mature miR guides AGO and TNRC6 to a partially complementary binding site within the 3’ UTR of a target mRNA. Following AGO/TNRC6 binding, decapping and deadenylase complexes are recruited on the target mRNA. Mammalian genomes encode for three highly regulated TNRC6 paralogues, GW182/TNRC6A, TNRC6B, and TNRC6C [13,14], which are functionally redundant [15]. Upon deadenylation and decapping, mRNAs become s substrate for cellular exonucleases, and are rapidly degraded (reviewed in [16]). As mammalian TNRC6 paralogues lack catalytic activity, it is conceivable that they do not play any “solo” roles in the cell. W indicates key tryptophan residues on TNRC6 for binding to AGO.

**Figure 2 genes-13-00414-f002:**
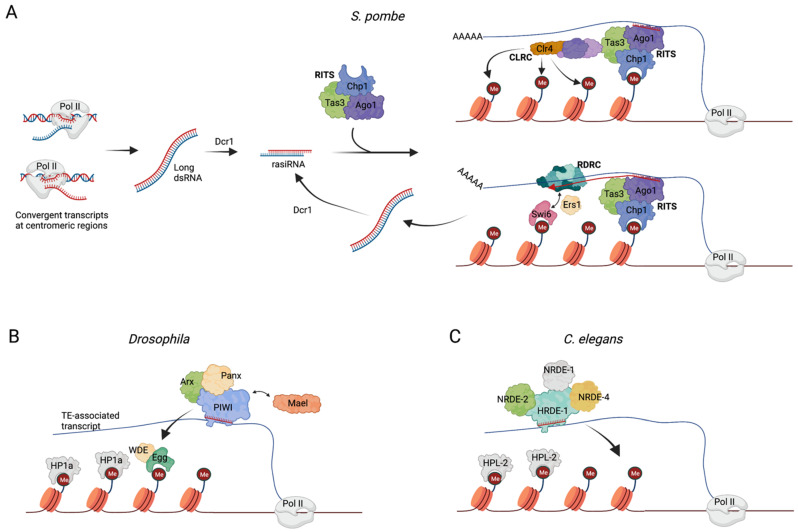
(**A**) The RITS complex in *S. pombe*. Convergent transcription at the targeted centromeric regions generates long double strand RNAs which are cleaved by Dicer1 (Dcr1) into small 25–30 nt dsRNAs, named repeat-associated siRNAs (rasiRNAs). One strand of the rasiRNA is loaded onto the RITS, and the resulting ribonucleoprotein complex is recruited on nascent transcripts spanning the targeted centromeric region. On the nascent transcript, the RITS can associate both with the CLRC complex to induce histone methylation, and with the RDRC complex to produce additional targeting rasiRNAs. RDRC recruitment is facilitated by Ers1, which acts as a bridge between the RDRC subunit Hrr1 and the chromodomain protein Swi6 [38]. (**B**) In *Drosophila*, Piwi–piRNA complexes bind TE-associated nascent transcripts, a process in which the proteins Arx and Panx take part. The histone methyltransferase, Egg, and the accessory protein, Wde, deposit H3K9me2/3 marks on transposon sequences, a process assisted by Mael. Next, HP1a binds to the targeted locus to maintain gene repression. (**C**) Mechanism of TGS in *C. elegans* operated by HRDE-1/WAGO-9 protein (details described in the main text).

**Figure 3 genes-13-00414-f003:**
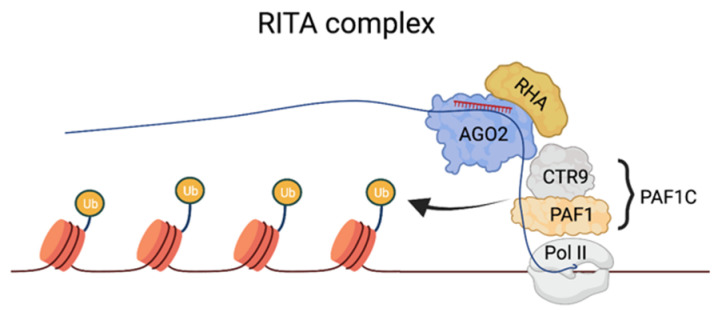
The RITA complex and the mechanisms of TGA at the p21 promoter. Binding sites on a nascent transcript overlapping the target are recognized by the saRNA–AGO2 complex. saRNA–AGO2 serves as a docking platform for components of PAF1C, which stimulate transcription initiation and recruit histone-modifying factors, such as E2/E3 ubiquitin ligase, to the Pol II complex to mediate H2B ubiquitination.

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
