# Peer review of "Roles of the Core Components of the Mammalian miRISC in Chromatin Biology"

_genes, 2022, doi:10.3390/genes13030414_

Round 1
Reviewer 1 Report
Here, the authors describe the role of microRNA-induced silencing complex components (specifically Argonaute and TNRC6) in mammalian chromatin regulation. miRISC is best understood as a mediator of post-transcriptional gene silencing in the cytoplasm. However, multiple, sometimes conflicting reports have described other roles for miRISC-associated proteins in the nucleus. The mechanistic underpinnings for these observations, especially in mammals, remain unknown. This review outlines studies that implicate nuclear miRISC in chromatin modification and remodeling, with a specific focus on mammals. They begin by describing the better understood mechanisms for RISC-mediated heterochromatin control in yeast. The authors then highlight two main pathways for control in mammals: transcriptional gene silencing and transcriptional gene activation, and then describe reports of each, attempting to highlight potential mechanistic insights and gaps that require further investigation. They also highlight the few studies that demonstrate TNRC6 involvement in chromatin function. They then conclude by proposing that a tool they have recently developed could provide new inroads for tackling outstanding questions and suggesting that understanding the nuclear roles of miRISC could reveal novel therapeutic strategies for certain diseases.
While the authors have identified an interesting and debated gap in the field, the organization is somewhat confusing and reads largely as a list of summaries without clear, unifying conclusions. In addition, several references are missing, and most studies highlighted in the current version are from over 5 years ago. Lastly, a review has recently been published with a similar goal: Nazer et al., Molecular Cell, 2021. The current manuscript mainly focuses on mammalian systems and chromatin modification, while Nazer et al. describes additional organisms and other nuclear functions of Argonaute (i.e. splicing). Nazer et al. also focuses heavily on Argonautes, while the current manuscript attempts to describe roles for Ago-TNRC6 complexes (though, this point could do with more support, see below). Nevertheless, with some reorganization, this review could be another welcome contribution to help organize the often-conflicting reports on this topic. I have included suggestions for improvements here:
- The current organization makes it difficult to parse overall conclusions in each section. It would be helpful to synthesize the results from all studies discussed into a set of conclusions and gaps (perhaps as a summary paragraph at the end of each section). It is also difficult to keep track of when and where different RISC components are participating. A table outlining all reports of RISC-mediated TGS/TGA might be helpful (i.e. list of genes/loci identified along with small RNA (either exogenous or miRNA), dependence upon Ago and/or TNRC6, proposed mechanism—DNA methylation, histone modification, enhancer organization etc., nascent ncRNA necessity, and a list of associated citations). This table could be accompanied by a model figure, if possible (see comment 4).
- While the authors focus on mammalian studies, they begin by extensively describing the pombeRITS complex. RITS is the most molecularly understood mechanism for small RNA-mediated chromatin silencing. However, it would be helpful to describe more closely related organisms where roles for nuclear Ago have been reproducibly observed, including other metazoans (flies, worms), to provide better context for conservation. Moreover, TGS in animals is best understood in the piRNA pathway, where the PIWI clade of Argonautes direct silencing of transposons. Thus, it may be helpful to mention piRNA silencing as a known, Argonaute-driven chromatin remodeling pathway in mammals (which, interestingly, has recently been shown to regulate endogenous gene programs in the worm germline: Cornes et al., Dev Cell 2021).
- One of the unique contributions of this review is its focus on Ago:TNRC6 (miRISC), as opposed to just Ago. However, I believe changes in emphasis and organization could help with clarity. For example, the authors rely heavily on structural comparisons to RITS, claiming that similarities suggest that RITS and RISC may share functions (lines 191-203; 573-582). However, the similarities they cite simply demonstrate that GW-rich regions are conserved binding platforms for Ago proteins (El-Shami et al., Genes & Development 2007), while the tie to chromatin modification remains sparse. If the authors wish to emphasize this comparison, a more systematic description of structural similarities could be helpful (i.e. conservation between TNRC6 and Tas3, especially beyond the ABD) as well as comparison between heterochromatin mechanisms/complexes between the two organisms. Alternatively, they might describe other complexes with a similar molecular architecture, such as Arabidopsis Ago4 and KTF1 and the C-terminal domains of RNA PolIV subunits, which direct chromatin methylation and similarly contain a GW-rich Ago binding domain. Such descriptions could better support that this is a conserved molecular mechanism. Lastly, one might mention the recent annotation of TNRC6 binding partners earlier (Hicks et al., Cell Reports 2017) to implicate this protein as a versatile organizer of effector proteins.
- Current figures could better support the core conclusions of text. Figure 1, for example, is not essential for the focus of this review and could easily be combined with Figure 2, if kept. It may be more helpful to include figures for current models for TGS/TGA and chromatin remodeling mechanisms—especially given their complexity.
- In Section 10, describing roles for TNRC6 in chromatin modification, the authors do not cite Liu et al., Biochemistry 2018, where they show that TNRC6 is not necessary for TGS of the progesterone receptor, one of the better understood systems for exogenous small RNA-induced TGS. It would be helpful to summarize all the findings regarding TNRC6 dependence in these chromatin functions to better understand when and where it is involved—highlighting what might be a core challenge (but biologically interesting feature) of studying the many functions of Argonaute: that its versatility is built upon differential recruitment of diverse complexes via combinatorial networks of scaffolds and clients (building on comment 3 above).
- Other missing citations:
Allo et al, “Argonaute-1 binds transcriptional enhancers and controls constitutive and alternative splicing in human cells,” PNAS 2014
Ameyar-Zazoua et al., “Argonaute proteins couple chromatin silencing to alternative splicing,” NSMB2012
Bai et al, “MicroRNA 195-5’ targets Foxo3 promoter region to regulate its expression in granulosa cells,” Int. J. Mol. Sci. 2021
Benhamed et al., “Senescence is an endogenous trigger for microRNA-directed transcriptional gene silencing in human cells,” Nature Cell Biology 2012
Chang et al., “Recruitment of KMT2C/MLL3 to DNA damage sites mediates DNA damage responses and regulates PARP inhibitor sensitivity in cancer,” Cancer Research 2021
Fallatah et al., “Ago1 controls myogenic differentiation by regulating eRNA-mediated CBP-guided epigenome reprogramming,” Cell Reports 2021: Ago1 is present in nucleus of C2C12 cells and associates with enhancers. Binds eRNAs and interacts with CBP of myotube cells. Depletion of Ago1 disrupts eRNA-CBP interaction, H3K27ac, and differentiation.
Gomez Acuna et al., “Nuclear role for human Argonaute-1 as an estrogen-dependent transcription coactivator,” JCB 2020: hAgo1 associates with estrogen receptor alpha enhancers, which is regulated by estrogen levels. hAgo1 depletion decreases long-range contacts between these enhancers and target promoters.
Griffin et al., “A nuclear role for the Argonaute protein AGO2 in mammalian gametogenesis,” BioRxiv, 2021: Mouse Ago2 nuclear roles in spermatogenesis =. Show that Ago2 binds nuclear RNAs and seems to associate preferentially with regions of open chromatin in meiotic cells. Association appears to be nascent transcript independent.
Huang et al., “Ago1 interacts with RNA polymerase II and Binds to Promoters of Actively Transcribed Genes in Human Cancer Cells,” PLOS Genetics 2013: Ago1, not Ago2, is associated with promoters of transcriptionally active genes in prostate cancer cells.
Liu et al., “Expression of TNRC6 (GW182) proteins is not necessary for gene silencing by fully complementary RNA duplexes,” Nucleic Acid Therapeutics 2019: shows that in HCT116 cells, localization of Ago2 is not dependent upon TNRC6.
Skourti-Stathaki et al., “R-loops induce repressive chromatin marks over mammalian gene terminantors,” Nature 2014: R-loops lead to antisense transcription that recruits Ago1/2 and leads to increased H2K9me2 marks.
Tarallo et al., “The nuclear receptor ERB engages AGO2 in regulation of gene transcription, RNA splicing and RISC loading,” Genome Biology 2017
Voutila et al., “Development and mechanism of small activating RNA targeting CEBPA, a novel therapeutic in clinical trials for liver cancer,” Molecular Therapy 2017: citation for CEBPA small RNAs in clinical trial (line 596-600).
Xiao et al., “Pervasive Chromatin-RNA Binding Protein Interactions Enable RNA-Based Regulation of Transcription,” Cell 2019: RNA Binding Protein ChIP-seq shows RBp interactions in active chromatin regions. Shows Ago2 enriched at transcribed regions in HepG2 cells.
Minor/line edits:
- A potentially interesting facet that the authors do not mention is that both miRISC and heterochromatin have been shown to organize using features that drive phase separation (reviewed in Larson and Narlikar, Biochemistry 2018). Do the authors think that there may be an intersection between these shared assembly principles that could impact miRISC function in heterochromatin formation?
- Current evidence points to differences in nuclear function between hAgo1 and hAgo2. While the authors do make sure to specify paralogs in text, it might be helpful to be more explicit and summarize any trends that might imply specialized functions between different Argonautes.
- The authors do not extensively describe the role of miRISC in regulating splicing, which may have been due to their focus on chromatin biology. However, chromatin context is often tied to changes in alternative splicing and so more discussion may be warranted (Allo et al., CSHS Quant Biol 2010).
- In the conclusions section, the authors highlight their own work as a strategy to study roles of nuclear miRISC. While an exciting avenue for probing the role of Ago-TNRC6 interactions in nuclear function, are there other experimental gaps that the authors see that can be filled? Overall, I think more of an emphasis on the specific challenges in the field would be helpful here.
- “Mammalians” should often be replaced with “mammals”
- Line 130-134: The authors state that since Ago mutants that are impaired in loading cannot retain TNRC6 in the cytoplasm (as demonstrated in Schraivogel et al, 2015) that target association is required for retainment of Ago-TNRC6 in the cytoplasm. It has been shown that so-called “empty” Argonautes are unstable and targeted for autophagic degradation (Kobayashi et al, Cell Reports2019). Therefore, alternative mechanisms might lead to the observed effect with these mutants (i.e. Ago is being targeted for turnover and no longer associating with TNRC6, thereby affecting its localization).
- Figure 2: TNRC6 is shown as a globular protein, while it is primarily intrinsically disordered with several folded domains. Thus, the representation is somewhat misleading.
- Color consistency between figures for Argonaute protein would help with clarity.
- A figure showing the domain architecture of different scaffold proteins (TNRC6 vs. Tas3, for example) might be helpful.
- Line 517: First use of term GW182. Either do not use this term or introduce protein as GW182/TNRC6 early on.
- Chromatin remodeling section could benefit from a brief introduction of chromatin organization, such as enhancers, insulators, and topological structures.
Author Response
Reviewer 1’s comments:
1. The current organization makes it difficult to parse overall conclusions in each section. It would be helpful to synthesize the results from all studies discussed into a set of conclusions and gaps (perhaps as a summary paragraph at the end of each section). It is also difficult to keep track of when and where different RISC components are participating. A table outlining all reports of RISC-mediated TGS/TGA might be helpful (i.e. list of genes/loci identified along with small RNA (either exogenous or miRNA), dependence upon Ago and/or TNRC6, proposed mechanism—DNA methylation, histone modification, enhancer organization etc., nascent ncRNA necessity, and a list of associated citations). This table could be accompanied by a model figure, if possible (see comment 4).
Response:
We agree with Reviewer #1 on this crucial point, and have remodeled the whole manuscript with the aim to allow the reader to easily keep track of the main findings described in the main text. While the description of the main findings has been kept for each of the studies described in the main text, we have included tables 1 and 2 to summarize studies reporting TGS and TGA mechanisms in mammals.
2. While the authors focus on mammalian studies, they begin by extensively describing the pombe RITS complex. RITS is the most molecularly understood mechanism for small RNA-mediated chromatin silencing. However, it would be helpful to describe more closely related organisms where roles for nuclear Ago have been reproducibly observed, including other metazoans (flies, worms), to provide better context for conservation. Moreover, TGS in animals is best understood in the piRNA pathway, where the PIWI clade of Argonautes direct silencing of transposons. Thus, it may be helpful to mention piRNA silencing as a known, Argonaute-driven chromatin remodeling pathway in mammals (which, interestingly, has recently been shown to regulate endogenous gene programs in the worm germline: Cornes et al., Dev Cell 2021).
Response:
In the main text we have now included a description of the piRNA pathway and referred to mechanisms found in Drosophila and C. elegans, to give context for evolutionary conservation, as suggested by Reviewer #1. Moreover, Figure 2 now contains additional cartoons describing TGS in these two species. We have also briefly described the main findings by Cornes et al. to provide a perspective of a wider involvement of the piRNA pathway in gene regulation in animals.
3. One of the unique contributions of this review is its focus on Ago:TNRC6 (miRISC), as opposed to just Ago. However, I believe changes in emphasis and organization could help with clarity. For example, the authors rely heavily on structural comparisons to RITS, claiming that similarities suggest that RITS and RISC may share functions (lines 191-203; 573-582). However, the similarities they cite simply demonstrate that GW-rich regions are conserved binding platforms for Ago proteins (El-Shami et al., Genes & Development 2007), while the tie to chromatin modification remains sparse. If the authors wish to emphasize this comparison, a more systematic description of structural similarities could be helpful (i.e. conservation between TNRC6 and Tas3, especially beyond the ABD) as well as comparison between heterochromatin mechanisms/complexes between the two organisms. Alternatively, they might describe other complexes with a similar molecular architecture, such as Arabidopsis Ago4 and KTF1 and the C-terminal domains of RNA PolIV subunits, which direct chromatin methylation and similarly contain a GW-rich Ago binding domain. Such descriptions could better support that this is a conserved molecular mechanism. Lastly, one might mention the recent annotation of TNRC6 binding partners earlier (Hicks et al., Cell Reports 2017) to implicate this protein as a versatile organizer of effector proteins.
Response:
We have remodeled the whole manuscript with new organization of sessions and titles to better guide the reader through the focus of the manuscript. For clarity, we have removed the part of the original manuscript that attempted to make a parallelism between the structural similarities between the RITS and the miRISC. We believe this information is not crucial for the sake of the main message of this review, after all.
The studies by Hicks at al. mentioned by Reviewer #1 are commented in section 7.
4. Current figures could better support the core conclusions of text. Figure 1, for example, is not essential for the focus of this review and could easily be combined with Figure 2, if kept. It may be more helpful to include figures for current models for TGS/TGA and chromatin remodeling mechanisms—especially given their complexity.
Response:
Figure 1 now includes only the general structure of the miRISC, and we have eliminated the representation of the miRNA biogenesis pathway, which, we agree with the Reviewer #1, was not crucial in this context.
As described in the main text, each gene under TGS and TGA has its own peculiarities in terms of co-factor involved, making the creation of summary cartoons describing these processes somewhat challenging. However, to attempt to satisfy Reviewer 1’s comment, we have now included, besides representations of the TGS in worm and fly (Figure 2), Figure 3 that represents TGA on the p21 promoter.
5. In Section 10, describing roles for TNRC6 in chromatin modification, the authors do not cite Liu et al., Biochemistry 2018, where they show that TNRC6 is not necessary for TGS of the progesterone receptor, one of the better understood systems for exogenous small RNA-induced TGS. It would be helpful to summarize all the findings regarding TNRC6 dependence in these chromatin functions to better understand when and where it is involved—highlighting what might be a core challenge (but biologically interesting feature) of studying the many functions of Argonaute: that its versatility is built upon differential recruitment of diverse complexes via combinatorial networks of scaffolds and clients (building on comment 3 above).
Response:
We have now commented on findings by Liu et al, 2018 (line 299). Moreover, as mentioned on response to comment 1, tables 1 and 2 have been added with the purpose to summarize the role of different RISC components, including TNRC6, in TGS and TGA.
6. Other missing citations:
Allo et al, “Argonaute-1 binds transcriptional enhancers and controls constitutive and alternative splicing in human cells,” PNAS 2014
Ameyar-Zazoua et al., “Argonaute proteins couple chromatin silencing to alternative splicing,” NSMB2012
Bai et al, “MicroRNA 195-5’ targets Foxo3 promoter region to regulate its expression in granulosa cells,” Int. J. Mol. Sci. 2021
Benhamed et al., “Senescence is an endogenous trigger for microRNA-directed transcriptional gene silencing in human cells,” Nature Cell Biology 2012
Chang et al., “Recruitment of KMT2C/MLL3 to DNA damage sites mediates DNA damage responses and regulates PARP inhibitor sensitivity in cancer,” Cancer Research 2021
Fallatah et al., “Ago1 controls myogenic differentiation by regulating eRNA-mediated CBP-guided epigenome reprogramming,” Cell Reports 2021: Ago1 is present in nucleus of C2C12 cells and associates with enhancers. Binds eRNAs and interacts with CBP of myotube cells. Depletion of Ago1 disrupts eRNA-CBP interaction, H3K27ac, and differentiation.
Gomez Acuna et al., “Nuclear role for human Argonaute-1 as an estrogen-dependent transcription coactivator,” JCB 2020: hAgo1 associates with estrogen receptor alpha enhancers, which is regulated by estrogen levels. hAgo1 depletion decreases long-range contacts between these enhancers and target promoters.
Griffin et al., “A nuclear role for the Argonaute protein AGO2 in mammalian gametogenesis,” BioRxiv, 2021: Mouse Ago2 nuclear roles in spermatogenesis =. Show that Ago2 binds nuclear RNAs and seems to associate preferentially with regions of open chromatin in meiotic cells. Association appears to be nascent transcript independent.
Huang et al., “Ago1 interacts with RNA polymerase II and Binds to Promoters of Actively Transcribed Genes in Human Cancer Cells,” PLOS Genetics 2013: Ago1, not Ago2, is associated with promoters of transcriptionally active genes in prostate cancer cells.
Liu et al., “Expression of TNRC6 (GW182) proteins is not necessary for gene silencing by fully complementary RNA duplexes,” Nucleic Acid Therapeutics 2019: shows that in HCT116 cells, localization of Ago2 is not dependent upon TNRC6.
Skourti-Stathaki et al., “R-loops induce repressive chromatin marks over mammalian gene terminantors,” Nature 2014: R-loops lead to antisense transcription that recruits Ago1/2 and leads to increased H2K9me2 marks.
Tarallo et al., “The nuclear receptor ERB engages AGO2 in regulation of gene transcription, RNA splicing and RISC loading,” Genome Biology 2017
Voutila et al., “Development and mechanism of small activating RNA targeting CEBPA, a novel therapeutic in clinical trials for liver cancer,” Molecular Therapy 2017: citation for CEBPA small RNAs in clinical trial (line 596-600).
Xiao et al., “Pervasive Chromatin-RNA Binding Protein Interactions Enable RNA-Based Regulation of Transcription,” Cell 2019: RNA Binding Protein ChIP-seq shows RBp interactions in active chromatin regions. Shows Ago2 enriched at transcribed regions in HepG2 cells.
Response:
We thank Reviewer #1 for pointing out important missing citations. We have now included a subgroup of those accordingly.
Minor/line edits:
1. A potentially interesting facet that the authors do not mention is that both miRISC and heterochromatin have been shown to organize using features that drive phase separation (reviewed in Larson and Narlikar, Biochemistry 2018). Do the authors think that there may be an intersection between these shared assembly principles that could impact miRISC function in heterochromatin formation?
Response:
This is a very intriguing aspect worth investigation. In the case of cytoplasmic miRISC, phase separation is determined by multivalent interactions between the glycine/tryptophan-rich domain of TNRC6 and cognate hydrophobic binding pockets within the Ago2 PIWI domain, as reported by Sheu-Gruttadauria et al, Cell 2018. Whether similar mechanisms take place in the formation of heterochromatin in the nucleus, maybe via interaction between AGO and TNRC6 paralogues is not known. It would be interesting to see if upon interruption of AGO-TNRC6 interaction in the nucleus (for example using a peptide-base strategy, La Rocca et al, eLife 2021) phase separation driven by heterochromatin is affected.
2. Current evidence points to differences in nuclear function between hAgo1 and hAgo2. While the authors do make sure to specify paralogs in text, it might be helpful to be more explicit and summarize any trends that might imply specialized functions between different Argonautes.
Response:
We agree with the Reviewer #1 that this is an emerging important aspect concerning the specific roles of the different AGO paralogues in the nucleus. However, as pointed out by Reviewer 1, the nuclear roles of AGO paralogues is the focus of a recent review by Nazar et al, to which we have directed the reader for further information (line 99).
3. The authors do not extensively describe the role of miRISC in regulating splicing, which may have been due to their focus on chromatin biology. However, chromatin context is often tied to changes in alternative splicing and so more discussion may be warranted (Allo et al., CSHS Quant Biol 2010).
Response:
We have now included a paragraph in section 6 where the role miRISC in regulating splicing via histone modification is discussed.
4. In the conclusions section, the authors highlight their own work as a strategy to study roles of nuclear miRISC. While an exciting avenue for probing the role of Ago-TNRC6 interactions in nuclear function, are there other experimental gaps that the authors see that can be filled? Overall, I think more of an emphasis on the specific challenges in the field would be helpful here.
Response:
We hope the revised version of the manuscript will satisfy this aspect brought up by Reviewer #1. Indeed, in this new version we have tried to emphasize some observational inconsistencies amongst the studies. Together with the questions listed in the discussion, we hope that the areas of future research will be clearly identified by the reader.
5. “Mammalians” should often be replaced with “mammals”
Response:
The term has been replaced throughout the main text, when appropriate.
6. Line 130-134: The authors state that since Ago mutants that are impaired in loading cannot retain TNRC6 in the cytoplasm (as demonstrated in Schraivogel et al, 2015) that target association is required for retainment of Ago-TNRC6 in the cytoplasm. It has been shown that so-called “empty” Argonautes are unstable and targeted for autophagic degradation (Kobayashi et al, Cell Reports2019). Therefore, alternative mechanisms might lead to the observed effect with these mutants (i.e. Ago is being targeted for turnover and no longer associating with TNRC6, thereby affecting its localization).
Response:
We thank the reviewer for pointing this. However, as shown in the Figure 6 of the original study by Schraivogel et al, the almost-complete loss of cytoplasmic signal of TNRC6 upon expression of loading-deficient AGO cannot be explained by a sole decrease of AGO levels. In fact, the expression levels of mutant AGO2 are only slightly lower than the WT counterpart (around 30% less). Nevertheless, we have removed the description of this experiment, to avoid any confusion.
7. Figure 2: TNRC6 is shown as a globular protein, while it is primarily intrinsically disordered with several folded domains. Thus, the representation is somewhat misleading.
Response:
Figure has been modified accordingly
8. Color consistency between figures for Argonaute protein would help with clarity.
Response:
Figures have been modified accordingly
9. A figure showing the domain architecture of different scaffold proteins (TNRC6 vs. Tas3, for example) might be helpful.
Response:
As pointed out in response to comment 3, we have removed the part of the original manuscript that attempted to make a parallelism between the structural similarities between the RITS and the miRISC. We hope Reviewer #1 will agree with this decision.
10. Line 517: First use of term GW182. Either do not use this term or introduce protein as GW182/TNRC6 early on.
Response:
Nomenclature has been corrected accordingly
11. Chromatin remodeling section could benefit from a brief introduction of chromatin organization, such as enhancers, insulators, and topological structures.
Response:
A brief introduction on chromatin organization, supported by a reference, has been added to the section
Reviewer 2 Report
In this review article, La Roca and Cavalieri describe the role of RISC components, mainly AGOs and TNRC6 in transcriptional gene silencing in mammals. They summarize recent developments in the understanding of the mechanism of TGS including via deposition of repressive histone marks guided by AGO and involvement of both AGO and TNRC6 in chromatin modification. The review is very detailed and covers an important field of research. The article is nicely written, and I enjoyed reading the review.
Considering the review is heavily focused on mammalian TGS, it might be useful for a broader or non-specialist audience to also highlight the advances made in the field regarding small RNA pathways/AGOs in TGS and histone modification in other model organisms like plants, worms etc. Authors can include either a small paragraph after the paragraph on yeast or redirect the readers to relevant review articles in the other model systems.
Minor comments
The authors should correct the use of the word mammals throughout the article. When used as a noun it should be mammal or mammals and only when used as an adjective it should be mammalian.
There are some previous editing strikethroughs of some words still remaining in the text which should be deleted.
Author Response
Reviewer #2’s comments:
Considering the review is heavily focused on mammalian TGS, it might be useful for a broader or non-specialist audience to also highlight the advances made in the field regarding small RNA pathways/AGOs in TGS and histone modification in other model organisms like plants, worms etc. Authors can include either a small paragraph after the paragraph on yeast or redirect the readers to relevant review articles in the other model systems.
Response:
We thank Reviewer #2 for pointing out this important aspect. We have now described sRNA-driven TGS mechanisms in lower animals and included relative figures summarizing them. We have also briefly commented on plants mechanisms.
Minor comments
The authors should correct the use of the word mammals throughout the article. When used as a noun it should be mammal or mammals and only when used as an adjective it should be mammalian.
Response:
Corrections have been made accordingly
There are some previous editing strikethroughs of some words still remaining in the text which should be deleted.
Response:
Corrections have been made accordingly